# Beta-Amyloid Instigates Dysfunction of Mitochondria in Cardiac Cells

**DOI:** 10.3390/cells11030373

**Published:** 2022-01-22

**Authors:** Sehwan Jang, Xavier R. Chapa-Dubocq, Rebecca M. Parodi-Rullán, Silvia Fossati, Sabzali Javadov

**Affiliations:** 1Department of Physiology, University of Puerto Rico School of Medicine, San Juan, PR 00936, USA; sehwan.jang@upr.edu (S.J.); xavier.chapa@upr.edu (X.R.C.-D.); 2Alzheimer’s Center at Temple, Lewis Katz School of Medicine, Temple University, Philadelphia, PA 19140, USA; rebecca.parodi-rullan@temple.edu (R.M.P.-R.); silvia.fossati@temple.edu (S.F.)

**Keywords:** Alzheimer’s disease, beta-amyloid, cardiomyocytes, coronary artery endothelial cells, mitochondria

## Abstract

Alzheimer’s disease (AD) includes the formation of extracellular deposits comprising aggregated β-amyloid (Aβ) fibers associated with oxidative stress, inflammation, mitochondrial abnormalities, and neuronal loss. There is an associative link between AD and cardiac diseases; however, the mechanisms underlying the potential role of AD, particularly Aβ in cardiac cells, remain unknown. Here, we investigated the role of mitochondria in mediating the effects of Aβ_1-40_ and Aβ_1-42_ in cultured cardiomyocytes and primary coronary endothelial cells. Our results demonstrated that Aβ_1-40_ and Aβ_1-42_ are differently accumulated in cardiomyocytes and coronary endothelial cells. Aβ_1-42_ had more adverse effects than Aβ_1-40_ on cell viability and mitochondrial function in both types of cells. Mitochondrial and cellular ROS were significantly increased, whereas mitochondrial membrane potential and calcium retention capacity decreased in both types of cells in response to Aβ_1-42_. Mitochondrial dysfunction induced by Aβ was associated with apoptosis of the cells. The effects of Aβ_1-42_ on mitochondria and cell death were more evident in coronary endothelial cells. In addition, Aβ_1-40_ and Aβ_1-42_ significantly increased Ca^2+^ -induced swelling in mitochondria isolated from the intact rat hearts. In conclusion, this study demonstrates the toxic effects of Aβ on cell survival and mitochondria function in cardiac cells.

## 1. Introduction

Alzheimer’s disease (AD) is a progressive neurodegenerative disorder that is expected to affect 13 million individuals, mostly older adults, in the U.S. by 2050 [1]. The main pathological characteristics of AD are the formation of extracellular deposits comprising aggregated β-amyloid (Aβ) fibers and intracellular neurofibrillary tangles formed by hyperphosphorylated tau protein. These alterations are associated with the loss of synapses, mitochondrial structural and functional abnormalities, oxidative stress, inflammation, and neuronal loss. A growing body of experimental and clinical studies demonstrate an associative link between AD and cardiac diseases such as heart failure, ischemic heart disease, and atrial fibrillation [2]. It has been proposed that cardiac dysfunction leads to cerebral hypoperfusion (hypoxia) associated with brain oxidative stress and acidosis that, in combination with Aβ aggregation, provoke neuronal degradation and progression of AD [3]. Oxidative stress can induce further production of Aβ and tau protein, another critical component involved in the pathogenesis of AD [4]. These studies suggest a feedback loop embedded in the crosstalk between oxidative stress and Aβ aggregation that stimulates the development and progression of AD. Although a causal role of cardiac dysfunction in AD progression can be explained, at least partially, by oxidative stress induced by decreased cerebral blood flow (hypoperfusion), it remains unclear whether AD *per se* can increase the risk for developing cardiac abnormalities.

Protein misfolding plays a central role in the pathogenesis of AD, and recent studies identified Aβ aggregates in the heart of patients with heart failure and dilated cardiomyopathy [5,6,7]. It has been suggested that Aβ oligomers may compromise cardiac function and lead to cardiovascular disease, the second leading cause of death (after pneumonia) in patients with AD [8]. Aβ_1-40_ and Aβ_1-42_, structurally similar to those found in the brain and similar amounts in the heart of AD patients, are associated with diastolic cardiac dysfunction [6]. Moreover, AD shares similar risk factors and common genetic profiles with cardiac diseases [5,9]. Mutations in *PSEN1* and *PSEN2,* associated with familial AD, and the apolipoprotein E ε4 allele, a risk factor for AD, were found in patients with dilated cardiomyopathy [5,10]. AD patients lacking symptomatic cardiovascular diseases exhibit cardiac dysfunction, including increased diastolic dysfunction and left ventricular wall thickness [6,11,12,13], as well as electrocardiographic abnormalities [12]. In patients with AD, Aβ_1-40_ and Aβ_1-42_ aggregates were found in the cardiomyocytes and interstitial spaces associated with cardiac dysfunction [6]. Plaque-like amyloid deposits observed in cardiomyopathy [5] could induce proteotoxicity and cell death in the heart [14]. However, despite a broad range of studies, the causal role of AD and amyloidosis in the development of cardiac abnormalities remains unknown. In addition, the source of the Aβ protein aggregates in the hearts of patients with AD is not entirely understood. However, high levels of Aβ_1-40_ oligomers have been detected in the cerebrospinal fluid of AD patients [15], and an increase in Aβ oligomers in peripheral plasma is now starting to be considered as a potential biomarker for AD [16]. Therefore, there is a potential for these toxic aggregation species to reach the heart. Although it is not yet clear how AD affects cardiac function, patients with AD are at increased risk for developing cardiac abnormalities.

The mechanisms of cell damage induced by Aβ are still subject to investigation; several lines of evidence demonstrate toxic intracellular effects of Aβ and suggest a role of the mitochondria in mediating these effects (reviewed in [17,18,19]). Aβ was found in the mitochondria of patients with AD [4,20] and increased Aβ aggregation was associated with mitochondrial dysfunction [21]. Together with oxidative stress and altered Ca^2+^ homeostasis, mitochondrial abnormalities have been shown to mediate neuronal dysfunction and cell death in AD patients and animal models of Aβ and tau pathology [22,23]. Mitochondrial dysfunction also mediates cerebral microvascular endothelial cell death induced by Aβ oligomers [21,24,25]. Mitochondria-mediated apoptosis can also be initiated by the effects of Aβ oligomeric species on membrane death receptors, such as DR4 and DR5, which trigger the extrinsic, and consequently the intrinsic, apoptotic pathways [26]. However, whether mitochondrial alterations act as a causal event and contribute to the etiology of AD or result from the disease pathology remains unknown. Notably, most of the studies elucidating the effect of Aβ on mitochondria have been conducted on brain tissue and neuronal or other brain cells, and thus, the mechanisms of Aβ production and its possible effects on cardiac cells remain unknown.

In this study, we elucidated the role of mitochondria in mediating the effects of Aβ_1-40_ and Aβ_1-42_ on cultured cardiomyocytes, coronary endothelial cells, and isolated cardiac mitochondria to clarify the potential capability of Aβ to impact mitochondrial bioenergetics and induce cardiac cell death. Our results demonstrated that Aβ_1-40_ and Aβ_1-42_ differentially affected cell viability and mitochondrial function; Aβ, mostly Aβ_1-42_ was found aggregated and instigated mitochondrial dysfunction associated with increased mitochondrial ROS (mtROS) and swelling, and reduced mitochondrial membrane potential (ΔΨ_m_), and calcium retention capacity (CRC).

## 2. Materials and Methods

### 2.1. Animals

Adult Sprague Dawley male rats (275–325 g) were purchased from Taconic (Hillside, NJ, USA). All experiments were performed according to protocols approved by the UPR Medical Sciences Campus Institutional Animal Care and Use Committee and conformed to the National Research Council Guide for the Care and Use of Laboratory Animals published by the US National Institutes of Health (2011, eighth edition).

### 2.2. Isolation of Mitochondria from Rat Hearts

Mitochondria were isolated from rat hearts by the method described previously [27]. Briefly, heart ventricles were cut and homogenized using a Polytron homogenizer in ice-cold sucrose buffer containing 300 mM sucrose, 20 mM Tris-HCl, and 2 mM EGTA, pH 7.2, and supplemented with 0.05% BSA. The heart homogenate was centrifuged at 2000× *g* for 3 min to remove cell debris. The supernatant was centrifuged at 10,000× *g* for 6 min to precipitate mitochondria and then washed again under the same conditions in the BSA-free sucrose buffer. The final pellet containing mitochondria was resuspended in the sucrose buffer.

### 2.3. Mitochondrial Permeability Transition Pore (mPTP) Opening

The swelling of mitochondria as an indicator of mPTP opening was determined in freshly isolated mitochondria (50 μg) by monitoring the decrease in light scattering at 525 nm in the presence or absence of Ca^2+^ [28]. The swelling buffer contained 125 mM KCl, 20 mM Tris-base, 2 mM KH_2_PO_4_, 1 mM MgCl_2_, 1 µM EGTA, 5 mM α-ketoglutarate, 5 mM L-malate, pH 7.1. Mitochondrial swelling was measured by adding Ca^2+^ to a total (accumulative) concentration of 100, 200, 300 µM with 5-min intervals at 37 °C using the Clariostar (BMG Labtech, Cary, NC, USA). The rates of swelling were calculated as decrements of absorbance values per minute (ΔA_525_·min^−1^·mg^−1^) and presented as a percentage of control.

### 2.4. Primary Primary Human Coronary Artery Endothelial Cells (HCAEC)

Primary HCAEC were cultured according to the manufacturer’s recommendations (ATCC). The cells were cultured in Vascular Cell Basal Media supplemented with VEGF-based Endothelial Cell Growth Kit (5 ng/mL rh VEGF, 5 ng/mL rh EGF, 5 ng/mL rh FGF, 15 ng/mL rh IGF-1, 10 mM L-glutamine, 0.75 units/mL heparin sulfate, 1 µg/mL hydrocortisone, 50 µg/mL ascorbic acid, 2% fetal bovine serum, ATCC) and 1% antibiotic solution (Sigma-Aldrich, Burlington, MA, USA). Cells were maintained in 95% air and 5% CO_2_ at 37 °C. Cells from passage 3–8 and less than 14 divisions were used, and less than 80% confluence was maintained during propagation.

### 2.5. H9c2 Cardiomyoblasts

H9c2 cardiomyoblasts were cultured according to the manufacturer’s recommendations (ATCC). Briefly, the cells were cultured in DMEM based modified media (4 mM L-glutamine, 4.5 g/L glucose, 1 mM sodium pyruvate, and 1.5 g/L sodium bicarbonate) supplemented with 10% fetal bovine serum and 1% antibiotic solution (Sigma-Aldrich) and maintained in 95% air and 5% CO_2_ at 37 °C. Cells maintained within 80–90% confluence from passages 3–10 were used in experiments. Mitochondrial bioenergetics, metabolism, and morphology of H9c2 cells are similar to primary cardiomyocytes [29].

### 2.6. Permeabilization of Cells

Cells were freshly harvested using trypsin-EDTA and then permeabilized for 10 min on ice in the buffer containing 300 mM sucrose, 10 mM Tris-HCl, 2 mM EGTA, pH 7.4, and 50 µg/mL saponin. After the permeabilization, cells were washed with equilibration buffer (100 mM sucrose, 10 mM Tris-HCl, 10 µM EGTA, pH 7.4), then dissolved in the incubation buffer (200 mM sucrose, 10 mM Tris-MOPS, 5 mM α-ketoglutarate, 2 mM malate, 1 mM P_i_, 10 μM EGTA-Tris, pH 7.4).

### 2.7. Mitochondrial CRC Assay

Freshly harvested and permeabilized cells were incubated at 37 °C in the 0.1 mL of the incubation buffer (200 mM sucrose, 10 mM Tris-MOPS, 5 mM α-ketoglutarate, 2 mM malate, 1 mM Pi, 10 μM EGTA-Tris, pH 7.4) containing 100 nM Calcium Green-5N. Calcium was added to increase matrix Ca^2+^ load, and the fluorescence intensity was recorded by the CLARIOStar microplate reader (BMG Labtech).

### 2.8. Cellular ROS Assay

Freshly harvested and permeabilized cells were incubated at 37 °C in the 0.1 mL of the incubation buffer (200 mM sucrose, 10 mM Tris-MOPS, 5 mM α-ketoglutarate, 2 mM malate, 1 mM Pi, 10 μM EGTA-Tris, pH 7.4) containing 50 mM sodium phosphate, pH 7.4, 50 µM Ampliflu^TM^ Red, 0.1 U/mL HRP (Sigma-Aldrich) and fluorescence signals were monitored using CLARIOStar microplate reader (BMG Labtech).

### 2.9. Fluorescence Immunocytochemistry

Cells were fixed and permeabilized with ice-cold methanol for 5 min. Fixed cells were washed with PBS 3 times, then blocked with 2% BSA in PBS for 30 min. Antibodies against Aβ (ab11132), ATP5A (ab176569), and calreticulin (ab92516) were used as per the manufacturer’s recommendation (Abcam, Waltham, MA, USA). The cells were incubated with the primary antibodies overnight at 4 °C, and then washed 3 times with PBS and incubated with Alexa Fluor 488 anti-mouse and/or Alexa Fluor 594 anti-rabbit secondary antibodies (Thermo Fisher, Waltham, MA, USA) for 1 h at room temperature. After incubation with secondary antibodies, the cells were washed 3 times with PBS, and 100 nM DAPI was added to visualize the nucleus. Images were captured by Olympus IX73 microscope with LUCPLFLN40X objective using Cellsense Dimension (Olympus, Center Valley, PA, USA) software. Image compositions were made using ImageJ.

### 2.10. Oligomerization and the Treatments of Aβ Peptides

The Aβ peptides Aβ_1-40_ and Aβ_1-42_ purchased from Sigma-Aldrich and Bon Opus Biosciences (Millburn, NJ, USA) were prepared as described previously [30]. First, lyophilized amyloid peptides were dissolved in 1,1,1,3,3,3-hexafluoro-2-propanol (HFIP) to 1 mM. After the peptides were completely monomerized, they were lyophilized and re-dissolved in DMSO to 10 mM, then were diluted to 1 mM by adding deionized water. The peptides were diluted in culture media to 0.2 mM and incubated at 4 °C for 48 h to form oligomers, which were then applied to cultured cells to a final concentration of 10 µM [31]. Low-binding surface plastic wares were used to prepare the amyloid peptides. To treat H9c2 cells, media was changed to serum-free media, Aβ peptides were added, and incubated 48–96 h. For primary HCAEC, Vascular Cell Basal Media supplemented with VEGF-based Endothelial Cell Growth Kit was used without modification. The HCAEC growth media was refreshed every 2 days in the presence or absence of Aβ peptides for 2–20 days.

### 2.11. Analysis of Cell Viability

Cell viability was determined by the AlamarBlue™ Cell Viability Assay Reagent (Thermo Fisher) as previously described [32].

### 2.12. Analysis of Cellular ATP, ΔΨm, and mtROS

Cells were live stained with 5 µM ATP-Red (Sigma-Aldrich), 10 µM JC-1 (Thermo Fisher), and 2 µM MitoSOX (Thermo Fisher) for quantification of cellular ATP levels, ΔΨ_m_, and mtROS production, respectively, following the manufacturers’ recommendation (Thermo Fisher). The fluorescence intensity of the dyes was measured using the CLARIOStar microplate reader (BMG Labtech). Fluorescence signals of ATP-Red and MitoSOX were normalized to the fluorescence intensity (blue signal) of the nucleus (Hoechst). JC-1 signals were presented as the red to green fluorescence intensity ratio of the dye.

### 2.13. Apoptosis Assay

For analysis of activated caspases 3 and 7, live cells were stained with 2 µM CellEvent Caspase-3/7 Green Detection Reagent (Thermo Fisher) in the presence of 50 nM Hoechst 33342 (Thermo Fisher) as per the manufacturer’s recommendation. Fluorescence signals were measured using the CLARIOstar microplate reader (BMG Labtech). The caspase fluorescence intensity was normalized to the fluorescence intensity (blue signal) of the nucleus (Hoechst).

### 2.14. Statistical Analysis

Data were analyzed using Student’s *t*-test. Results are presented as mean ± SEM. *p* < 0.05 was considered statistically significant. The number of biological samples but not technical replicates were used as a sample size.

## 3. Results

### 3.1. Aβ Decreased Cell Viability and Impaired Cell Morphology

The cell viability of HCAEC grown in the culture media containing 10 µM Aβ_1-40_ or Aβ_1-42_ for 20 days was significantly decreased in comparison with control cells. The cell viability of Aβ_1-40_ or Aβ_1-42_ groups were 8.6% and 24% lower (*p* < 0.05) than the control, respectively (Figure 1A). Morphological analysis using phase-contrast microscopy in Aβ_1-42_-treated cells revealed irregular-shaped lumps appearing bright compared to the control and Aβ_1-40_-treated cells (Figure 1B). Incubation of H9c2 cardiomyocytes with 10 µM Aβ_1-42_ for 96 h resulted in a 39% (*p* < 0.05) decrease of cell viability, whereas Aβ_1-40_ reduced the cell viability only by 8% (Figure 1C). H9c2 cells challenged with Aβ_1-42_ showed abnormally shrinking cell morphology, in addition to abnormal bright aggregates (Figure 1D).

### 3.2. Aggregated Aβ Accumulated Inside of Cells

To investigate the possible inclusion of Aβ aggregates in subcellular compartments, primary HCAEC and H9c2 cardiomyocytes incubated with Aβ_1-40_ or Aβ_1-42_ were visualized using immunocytochemistry. Control primary HCAEC showed only low perinuclear signal, which could indicate a low level of Aβ and/or background staining, whereas Aβ_1-40_-treated cells demonstrated the higher intensity of the cytoplasmic signal. Primary HCAEC challenged with Aβ_1-42_ showed a strong signal of irregular-shaped aggregates with a size of 1–50 µm (Figure 2A). Similar patterns of intracellular aggregation of Aβ were observed in H9c2 cells. The control group showed low perinuclear signal, whereas cells grown with Aβ_1-40_ showed more cytoplasmic signal (in addition to the nuclei) compared to the control. Cells grown with Aβ_1-42_ showed irregular-shaped aggregates in the cytoplasm (Figure 2B) that were absent in the control.

### 3.3. Aβ Induced Mitochondrial and Cellular Dysfunction

Next, we investigated the role of mitochondria in the cells exposed to Aβ. The primary HCAEC incubated with 10 µM Aβ_1-42_ for 20 days showed a 37% (*p* < 0.05) increase of mtROS, and a 179% (*p* < 0.01) increase of cellular (total) ROS, compared to the control cells (Figure 3A,B). Aβ_1-40_ and Aβ_1-42_ induced depolarization of the mitochondrial membrane as evidenced by decreased ΔΨm. The primary HCAEC treated with Aβ_1-40_ and Aβ_1-42_ showed a 20% (*p* < 0.05) and 41% (*p* < 0.01) decrease of ΔΨm, respectively, in comparison with the control (Figure 3C). Aβ_1-42_ induced a 2.4-fold (*p* < 0.01) increase of activated caspase 3/7, indicating increased apoptosis (Figure 3D). Interestingly, Aβ_1-40_ and Aβ_1-42_ increased ATP levels by 13% and 29% (*p* < 0.01), respectively (Figure 3E). Similar trends in cellular and mitochondrial parameters were observed in H9c2 cells. The cells incubated with 10 µM Aβ_1-42_ for 20 days showed a 22% (*p* < 0.01) increase of mtROS, and a 24% (*p* < 0.01) increase of cellular ROS, compared to the control cells (Figure 3F,G). Aβ_1-42_ decreased ΔΨ_m_ by 33% (*p* < 0.05) in H9c2 cells (Figure 3H), which showed 25% (*p* < 0.01) more activated caspase 3/7 than the control cells (Figure 3I). In H9c2 cells, only Aβ_1-42_ increased the ATP levels by 13% (*p* < 0.05), whereas Aβ_1-40_ had no significant effects (Figure 3J).

### 3.4. Cells and Mitochondria Demonstrated Early Response to Aβ

To elucidate the possible cause of the Aβ-induced cell death, we investigated the earlier changes that could predispose the cells and mitochondria to further functional alterations leading to cell death. Incubation of primary HCAEC with 10 µM Aβ_1-40_ or Aβ_1-42_ for 48 h did not decrease the cell viability compared to the control (Figure 4A). Analysis of cellular morphology revealed that Aβ_1-42_ increased dark, grainy aggregates in the cell cytoplasm that were not observed in the control and Aβ_1-40_-treated groups (Figure 4B). Aβ_1-42_ did not affect mtROS levels after 48 h, but it increased cellular ROS by 59% (*p* < 0.05) (Figure 4C,D). In addition, mitochondria were found depolarized in the cells incubated with Aβ_1-40_ or Aβ_1-42_ that demonstrated 27% (*p* < 0.05) and 45% (*p* < 0.05) fewer ΔΨ_m_ for Aβ_1-40_ and Aβ_1-42_, respectively, in comparison with the control (Figure 4E). No significant increase in caspase activation was observed in the presence of the Aβ (Figure 4F). Aβ_1-42_ increased ATP levels by 59% (*p* < 0.05) while Aβ_1-40_ did not affect ATP (Figure 4G). Likewise, the H9c2 cells incubated with 10 µM Aβ_1-40_ or Aβ_1-42_ for 48 h did not show any change in cell viability (Figure 4H).

Cardiomyocytes also showed dark, grainy aggregates in the cytoplasm of incubated with Aβ_1-42_ for 48 h. The control and Aβ_1-40_-treated groups did not show remarkable changes in morphology (Figure 4I). Aβ_1-42_ induced a 17% (*p* < 0.05) increase of mtROS and a 19% (*p* > 0.05) increase of cellular ROS after 48 h of incubation (Figure 4J,K). Aβ_1-42_ induced a 15% (*p* < 0.05) decrease of the ΔΨm (Figure 4L) and increased caspase 3/7 activation by 15% (*p* < 0.05) compared with the control (Figure 4M). Both Aβ_1-40_ and Aβ_1-42_ had no significant effect on the ATP levels in H9c2 cells (Figure 4N). Analysis of mitochondrial CRC demonstrated high sensitivity of the cells to Aβ_1-42_. Incubation of primary HCAEC with 10 µM Aβ_1-42_ for 48 h decreased the mitochondrial CRC by 15% (*p* < 0.05) compared to control cells (Figure 5A,B). Likewise, 10 µM Aβ_1-42_ induced a 20% (*p* < 0.05) decrease of the mitochondrial CRC in H9c2 cardioblasts (Figure 5C,D). Aβ_1-40_ did not affect the mitochondrial CRC in both types of cells.

Immunocytochemical analysis of the primary HCAEC incubated with 10 µM Aβ_1-42_ for 48 h showed aggregated Aβ throughout the cytoplasm (Figure 6A, green). Mitochondria in Aβ_1-42_-treated cells showed mostly fragmented mitochondria, whereas control cells and the cells incubated with Aβ_1-40_ presented a well-organized mitochondrial network (Figure 6A, white). Likewise, mitochondria were more fragmented in H9c2 cardioblasts incubated with 10 µM Aβ_1-42_ for 48 h (Figure 6B, white) and showed aggregated Aβ in the cytoplasm (Figure 6B, green).

### 3.5. Aβ induced Dysfunction of Cardiac Mitochondria In Vitro

To investigate the possible direct effects of Aβ on the mitochondria, we measured mitochondrial swelling rates, mtROS, ΔΨm, and ATP levels in mitochondria that were isolated from healthy rat hearts. The mitochondria were incubated with 10 µM Aβ_1-40_ or Aβ_1-42_ for 10 min. Results demonstrated that Aβ_1-40_ and Aβ_1-42_ further increased the Ca^2+^ -induced mitochondrial swelling rate by 34% (*p* < 0.05) and 97% (*p* < 0.05), respectively, in comparison with the control (Figure 7A,B). The swelling of mitochondria apparently was induced by mPTP opening since the addition of sanglifehrin A, a cyclophilin D inhibitor, completely prevented the mitochondrial swelling in control and Aβ-treated groups (Figure 7B). Aβ_1-42_ slightly (5%, *p* < 0.05) increased mtROS (Figure 7C), decreased the ΔΨm by 12% (*p* < 0.05, Figure 7D), and increased ATP levels (8%, *p* < 0.05), (Figure 7E). Aβ_1-40_ did not induce any remarkable changes in mtROS, ΔΨm, and ATP levels.

## 4. Discussion

Results of the present study demonstrate the direct detrimental effects of Aβ, particularly Aβ_1-42_, on cardiomyocytes and primary coronary endothelial cells. On the other hand, the cardiac cells demonstrated a different sensitivity to Aβ. Comparative analysis of early (48 h) responses for both cells revealed severe effects of Aβ_1-42_ on the mitochondria. We have previously shown that the mitochondrial bioenergetics, metabolism, function, and morphology of H9c2 are similar to primary cardiomyocytes [29].

Several metabolic alterations could be involved in the adverse action of Aβ_1-42_ on mitochondria. Aβ oligomers have been shown to disrupt cell membrane permeability and calcium homeostasis in neurons [33], cerebral endothelial cells [21,24,34], and cardiomyocytes [5]. Our results showed that Aβ decreased the cell viability of cardiac cells as well as primary HCAEC. Our study and others have reported that mitochondrial dysfunction and mitochondria-mediated apoptosis play a crucial role in the pathogenesis of both AD and heart failure [19,35]. However, despite these findings, it is not clear whether these changes are the cause or just another result of aging in AD and heart failure patients. Our study showed early signs of cellular and mitochondrial dysfunctions because of Aβ. In addition to cerebral vasculature, Aβ peptides were found in atherosclerotic lesions and platelets [36]. High blood levels of Aβ_1-40_ in patients with coronary heart disease were identified as a marker that could predict a high risk of mortality [37]. Analysis of the hearts and brains of patients with AD demonstrated that the hearts of several patients contain Aβ deposits (Aβ_1-40_ and Aβ_1-42_) that are structurally similar to those found in the brain. Similar amounts of Aβ_1-40_ and Aβ_1-42_ peptides were also found in the hearts and brains of AD patients. [6]. The patients with Aβ deposition in the heart presented diastolic dysfunction, although none of them had a history of coronary heart diseases. Aβ induced decreased complex activity, H_2_O_2_ production, ATP synthesis, state 3 and 4 respiration, and release of cytochrome c in rat muscle mitochondria [38]. Interestingly, similar mitochondrial dysfunction and apoptotic mechanisms were shown by our group in cerebral microvascular endothelial cells challenged with multiple Aβ variants, including Aβ_1-40_ and Aβ_1-42_ [19,21,24,26].

Our results demonstrated that Aβ increases mitochondrial or cellular ROS production, which, in turn, can induce further propagate Aβ and tau protein production [4]. Analysis of the hearts of AD patients resulted in the discovery of the presence of amyloid aggregates (Aβ_1-40_ and Aβ_1-42_) in cardiomyocytes and interstitial spaces, and that was associated with myocardial diastolic dysfunction [6]. Although it is not yet clear what is the source of the Aβ aggregates in the hearts of AD patients, accumulation of Aβ has been observed not only in the heart but also in other organs of AD patients [39], suggesting that circulating Aβ can be deposited or that these peptides can be produced in the heart, among other organs. Furthermore, circulating Aβ oligomers, recognized as the major toxic species for multiple cell types, could participate in the induction of cardiac or vascular endothelial dysfunction through oxidative stress mechanisms. On the other hand, amylin and amyloid deposits were found in the hearts of non-AD patients with diabetic and idiopathic cardiomyopathy [5,40].

Our data demonstrate that Aβ has a direct effect on isolated mitochondria, although Aβ is not produced in mitochondria [41]. Existing data on the transportation/localization of Aβ in mitochondria are still controversial. Accumulation of amyloid precursor protein across mitochondrial import channels (TOM and TIM) was detected in brain mitochondria of AD patients [42]. Alternatively, extracellular oligomeric Aβ aggregates may affect mitochondrial function by triggering cell membrane receptors (e.g., death receptors) and/or other signal transduction pathways that converge on the mitochondria [19]. Aβ has been shown by many studies, including ours, to diminish mitochondrial respiration and increase the levels of mtROS in multiple cell types, including neuronal and cerebral endothelial cells [19,21,24,25,34,43], and thus, can induce further Aβ and tau protein production [4]. However, there are few if any studies that investigate the direct effects of Aβ on cardiomyocytes and coronary endothelial cells.

Aβ neurotoxicity is associated with intraneuronal Ca^2+^ dyshomeostasis; increased cytosolic Ca^2+^ levels were detected in AD mice [44] and after application of soluble Aβ oligomers to the brain of wild-type mice [45]. Our results demonstrated that Aβ accelerated the mitochondria swelling caused by Ca^2+^ overload, an indicator of the mPTP opening. Studies on the brain tissue and neuronal cells revealed that the mPTP opening induced by high Ca^2+^ is one of the mechanisms that mediate the effects of Aβ and tau protein leading to mitochondrial dysfunction and cell death [46]. Our results showed that the cyclophilin D inhibitor sanglifehrin A completely inhibited mitochondrial swelling induced by Ca^2+^ and by Ca^2+^ + Aβ, suggesting the effect of Aβ on the swelling depends on the mPTP opening. Cyclophilin D, a major mPTP regulator in the matrix, was found increased in AD-affected brain regions; Aβ-cyclophilin D complex was detected in Aβ-rich mitochondria from AD brain and transgenic AD mice, suggesting that the effects of Aβ to induce mPTP opening are mediated through its interaction with cyclophilin D [47]. Conversely, genetic or pharmacological inhibition of cyclophilin D prevented Aβ-induced mPTP opening and cell death [48], decreased mitochondrial and neuronal perturbations, and improved learning and memory in AD [49].

Mitochondrial quality control mechanisms, including mitophagy and mitochondrial biogenesis and dynamics, are compromised by aging. A large number of factors and mechanisms maintain the structural and functional integrity of mitochondria, and modulation of their intensity/efficiency with aging apparently impairs structural and functional integrity of mitochondria and diminishes the mitochondrial quality control leading to mitochondrial dysfunction and, eventually, age-related diseases such as AD [50]. In this context, our results demonstrated that, among other factors, Aβ accumulation in cardiac cells with aging might play a certain role in mitochondrial/cellular dysfunction in the elderly population.

## 5. Conclusions

This study reveals adverse effects of Aβ, particularly Aβ_1-42_, on cardiomyocytes and coronary endothelial cells that could be mediated, among other mechanisms, through functional and metabolic alterations of mitochondria. Like neurons and cerebral endothelial cells, mitochondrial abnormalities might stimulate the intrinsic apoptotic pathway leading to cell death. Although the source of Aβ and mechanisms of accumulation of Aβ aggregates in the heart in the aged population and AD patients remains undiscovered, this study opens new perspectives for elucidating the potential role of Aβ in cardiac dysfunction.

## Figures and Tables

**Figure 1 cells-11-00373-f001:**
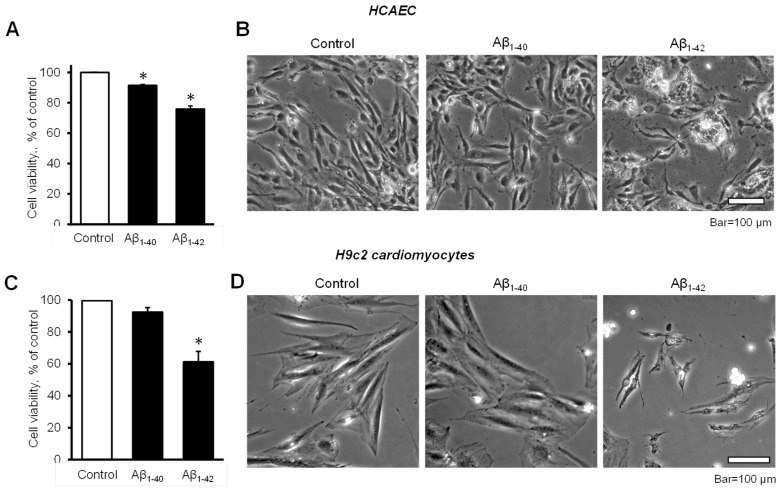
Aβ decreased cell viability and impaired cell morphology (**A**,**B**): Primary HCAEC grown for 20 days in the presence and absence of 10 µM Aβ_1-40_ or Aβ_1-42_. (**A**): Cell viability. *n* = 4 per group. * *p* < 0.05 vs. control. (**B**): Representative phase-contrast images of control (vehicle: DMSO) cells and cells grown with 10 μM Aβ_1-40_ or Aβ_1-42_. Bar = 100 µm (**C**,**D**): H9c2 cardiomyocytes grown with Aβ_1-40_ or Aβ_1-42_ for 96 h. (**C**): Cell viability. *n* = 4 per group. * *p* < 0.05 vs. control. (**D**): Representative images of control (vehicle: DMSO) cells and cells grown with 10 μM Aβ_1-40_ or Aβ_1-42_. Bar = 100 µm.

**Figure 2 cells-11-00373-f002:**
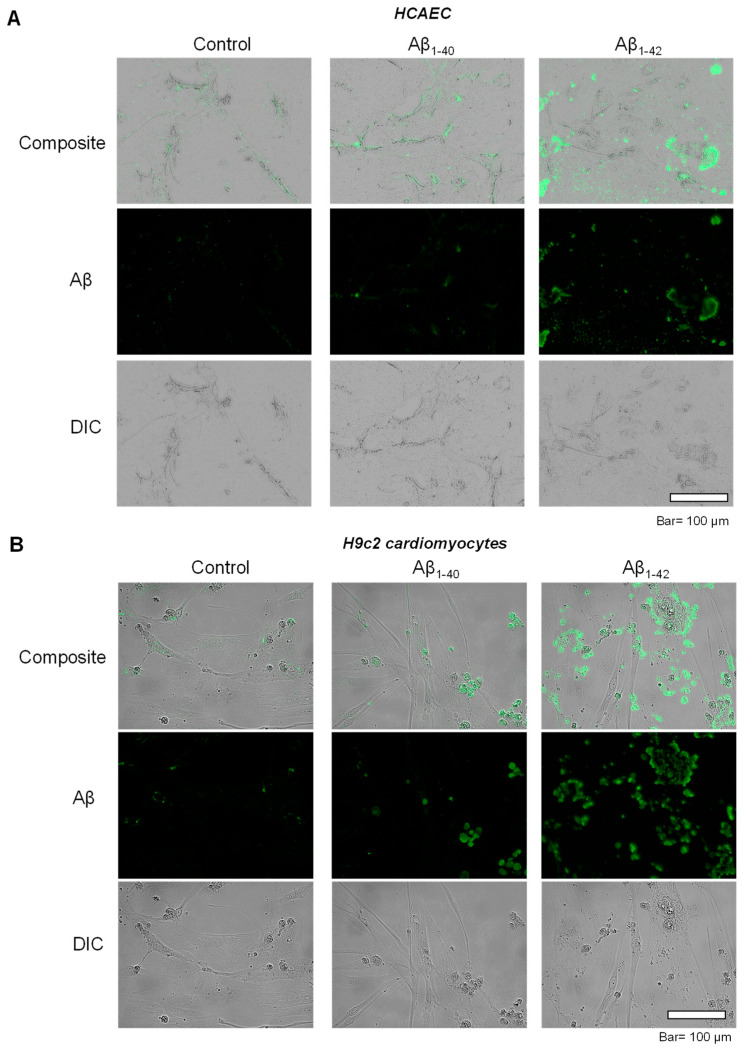
Accumulation of aggregated Aβ. (**A**): Immunostaining of Aβ in primary HCAEC grown with 10 µM Aβ_1-40_ or Aβ_1-42_ for 20 days. Bar= 100 µm (**B**): Immunostaining of Aβ in H9c2 cardiomyocytes grown with 10 µM Aβ_1-40_ or Aβ_1-42_ for 96 h. Bar = 100 µm.

**Figure 3 cells-11-00373-f003:**
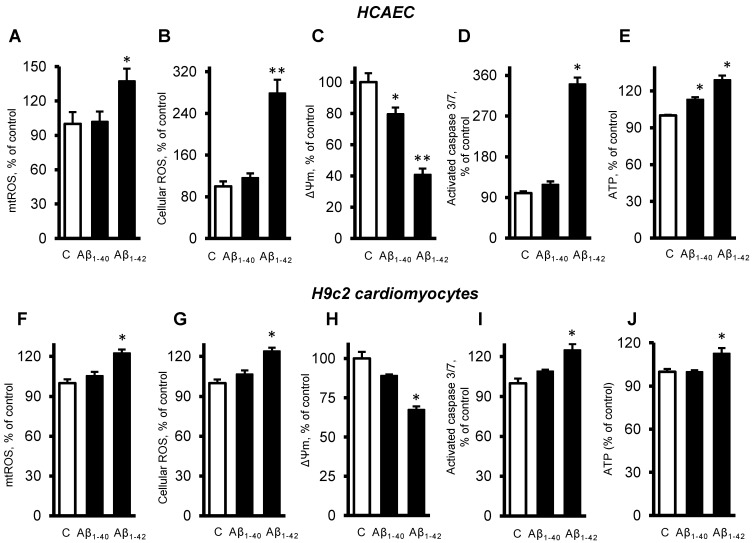
Aβ-induced mitochondrial and cellular dysfunction. (**A**–**E**): Primary HCAEC cultured for 20 days in the presence and absence of 10 µM Aβ_1-40_ or Aβ_1-42_. (**A**): mtROS levels measured by MitoSOX Red (Thermo Fisher). (**B**): Cellular ROS levels measured using Ampiflu Red (Sigma) in permeabilized cells. (**C**): ΔΨ_m_ measured by JC-1 (Thermo Fisher). (**D**): Activated caspase 3/7 levels measured using the CellEvent™ Caspase-3/7 Green Detection Reagent (Thermo Fisher). (**E**): ATP levels measured by ATP-Red (Sigma). (**F**–**J**): H9c2 cardiomyocytes cultured for 96 h in the presence or absence of 10 µM Aβ_1-40_/Aβ_1-42_. All parameters were measured by the same methods used for primary HCAEC (**A**–**E**). (**F**): mtROS levels. **G:** Cellular ROS levels. (**H**): ΔΨ_m_. (**I**): Activated caspase 3/7 levels. (**J**): ATP levels. *n* = 4 per group for all parameters of primary HCAEC and H9c2 cardiomyocytes. * *p* < 0.05, ** *p* < 0.01 vs. control (**C**).

**Figure 4 cells-11-00373-f004:**
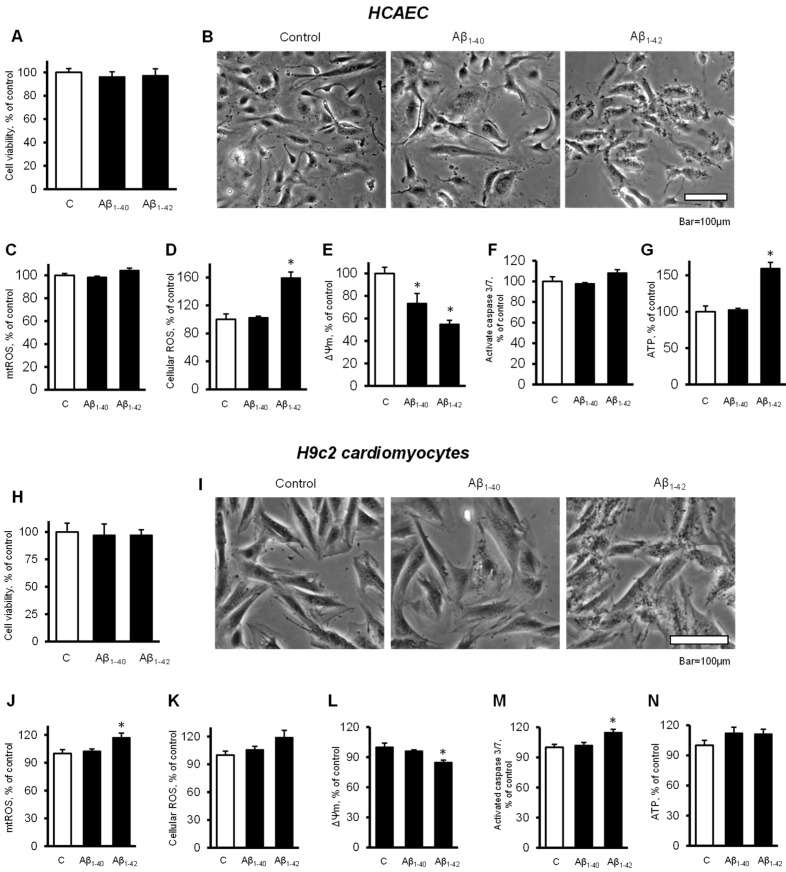
Cells and mitochondria demonstrated early response to Aβ. (**A**–**G**): Primary HCAEC grown for 48 h in the presence and absence of 10 µM Aβ_1-40_ or Aβ_1-42_. (**A**): Cell viability after 48 h of treatment. *n* = 4 per group. (**B**): Representative phase-contrast images of control (vehicle: DMSO) cells and cells treated with 10 μM Aβ_1-40_ or Aβ_1-42_ for 48 h. Bar = 100 µm (**C**): mtROS levels (**D**): Cellular ROS levels. (**E**): ΔΨ_m_. (**F**): Activated caspase 3/7 levels. (**G**): ATP levels. (**H**–**N**): H9c2 cardiomyocytes grown for 48 h in the presence and absence of 10 µM Aβ_1-40_ or Aβ_1-42_. (**H**): Cell viability after 48 h of treatment. (**I**): Representative phase-contrast images of cells. Bar = 100 µm (**J**): mtROS levels (**K**): Cellular ROS levels. (**L**): ΔΨ_m_. (**M**): Activated caspase 3/7 levels. (**N**): ATP levels. *n* = 4 per group for all parameters of primary HCAEC and H9c2 cardiomyocytes. * *p* < 0.05 vs. control (**C**).

**Figure 5 cells-11-00373-f005:**
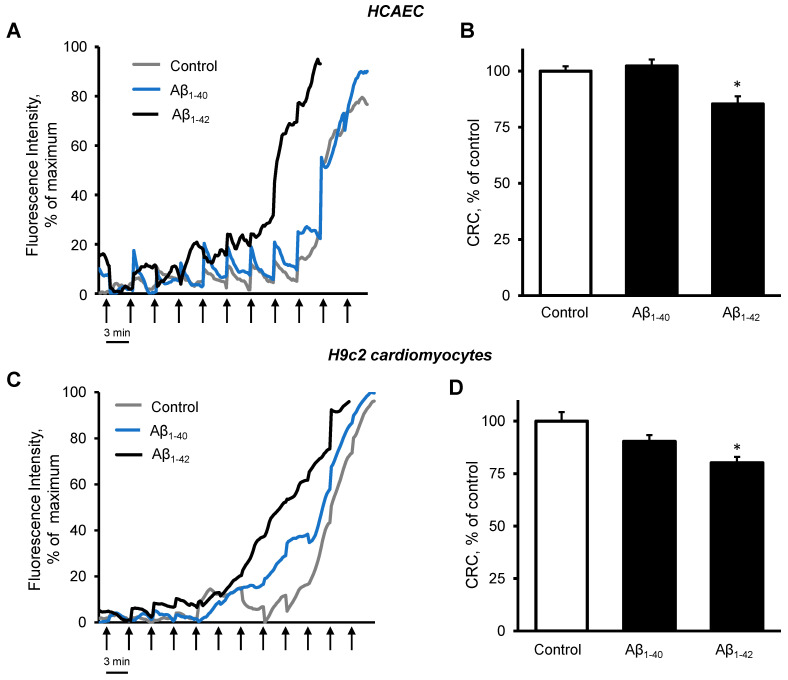
Aβ_1-42_ decreased CRC. (**A,B**): Primary HCAEC grown for 48 h in the presence and absence of 10 µM Aβ_1-40_ or Aβ_1-42_. (**A**): Representative traces of the fluorescence intensity of Calcium Green 5N. (**B**): Quantification of CRC calculated from the burst cycles (cycles that showed the highest fluorescence signal increase). (**C**,**D**): H9c2 cardiomyocytes were grown for 48 h in the presence and absence of 10 µM Aβ_1-40_ or Aβ_1-42_. (**C**): Representative traces of the fluorescence intensity of Calcium Green 5N. (**D**): CRC was calculated from the burst cycles. The addition of 1 nmol calcium was done every 3 min (one cycle) at 37 °C. Assays without cells were used as a baseline. *n* = 6 per group. * *p* < 0.05 vs. control.

**Figure 6 cells-11-00373-f006:**
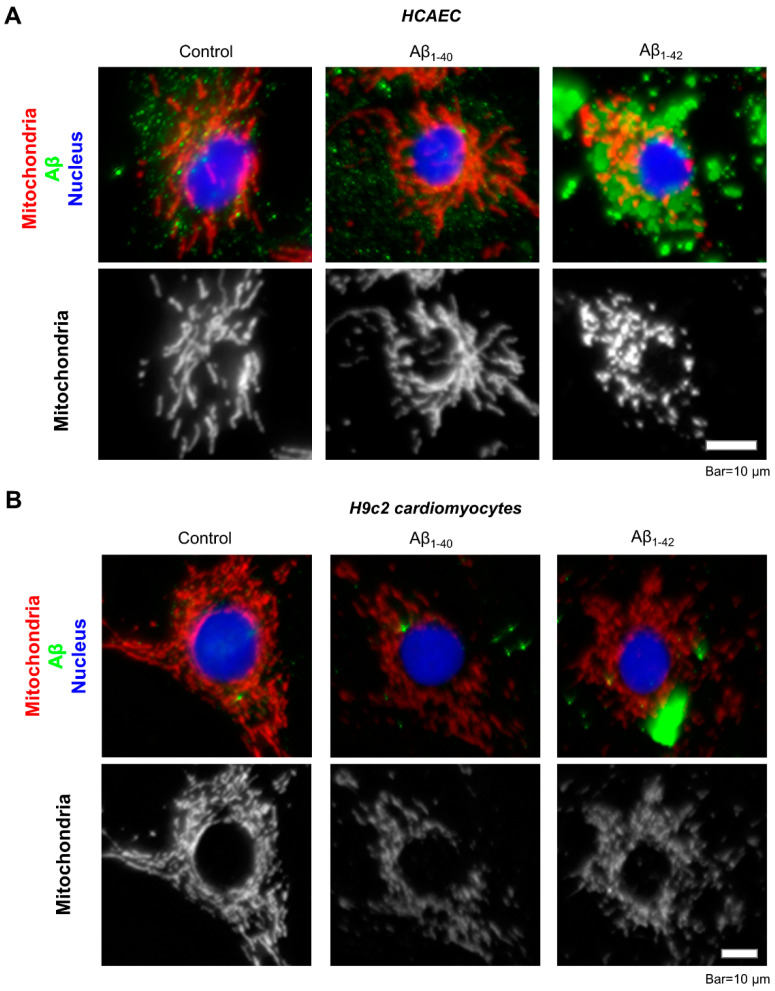
Visualization of Aβ aggregates and mitochondria by immunostaining. (**A**): Representative images of primary HCAEC. (**B**): Representative images of H9c2 cardiomyocytes. Primary HCAEC and H9c2 cardiomyocytes grown for 48 h in the presence and absence of 10 µM Aβ_1-40_ or Aβ_1-42_ were fixed and stained with anti-amyloid beta (green) and anti-ATP5A antibody (red). Nuclei were stained by DAPI. Bar = 10 µm.

**Figure 7 cells-11-00373-f007:**
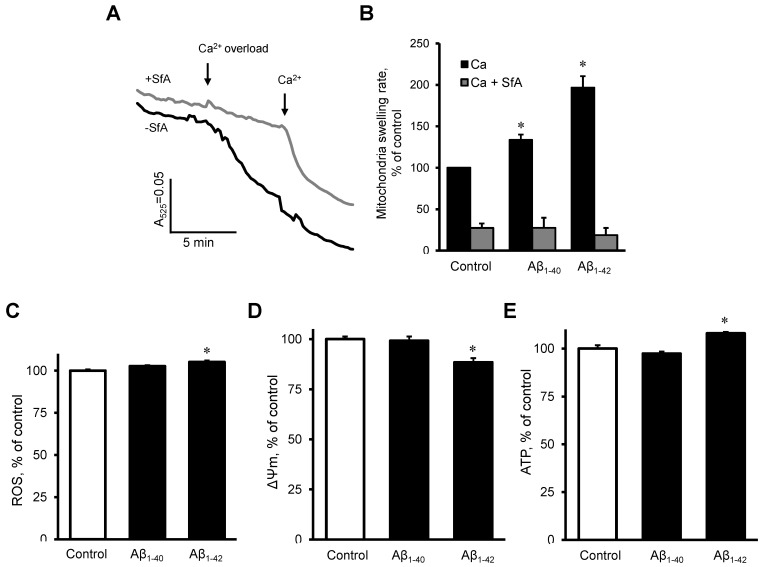
The effect of Aβ on isolated cardiac mitochondria. (**A**,**B**): Isolated rat heart mitochondria were used to analyze mitochondria swelling. (**A**): Mitochondrial swelling measured as decrement of A_525_ during the first 2 min of the calcium overload. Sanglifehrin A (SfA), a cyclophilin D (an mPTP regulator) inhibitor, was used to demonstrate that the swelling was mediated by mPTP. (**B**): Quantification of mitochondrial swelling rate presented as a percentile of control. Aβ_1-40_ or Aβ_1-42_ were added 10 min before the experiment to 10 µM. 0.1% DMSO was used as a vehicle. *n* = 3–6 per group. (**C**): mtROS levels (**D**): ΔΨ_m_. (**E**): ATP levels. *n* = 4 per group. * *p* < 0.05 vs. control.

## Data Availability

The data that support the findings of this study are available from the corresponding author upon reasonable request.

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
