# Peer review of "Beta-Amyloid Instigates Dysfunction of Mitochondria in Cardiac Cells"

_cells, 2022, doi:10.3390/cells11030373_

Round 1

Reviewer 1 Report

Reading the manuscript written by Jang et al., was really interesting. The article aims to evaluate the role of mitochondria in mediating the effects of Aβ1-40 and Aβ1-42 in cultured cardiomyocytes and primary coronary endothelial cells. From a preclinical perspective, this research reveals adverse effects of Aβ, particularly Aβ1-42 on cardiomyocytes and coronary endothelial cells that could be mediated, among other mechanisms, through functional and metabolic alterations of mitochondria. This manuscript is written in a concise and orderly manner. The methodologies are appropriate and aligned with the proposed objectives. The massage from this manuscript is quite meaningful. However, I have several comments:

Why have Alexa 160 Fluor 488 anti-mouse and/or Alexa Fluor 594 anti-rabbit been selected as secondary antibodies?

In the context of your results, how can you explain: “Many of mitochondrial regulation mechanisms become loose during aging. For example, mitochondrial biogenesis and mitophagy decline while mitochondria fission enhances along aging. It is possible that mild deficit in one or several of these mitochondrial regulation mechanisms determined by one’s genetic background could set the motion of a downward spiral and all of these mitochondrial deficits could come into play at some points that impairs mitochondrial integrity, which causes damage, affects repair and/or replication of mtDNA and thus accelerates the accumulation of mtDNA changes, leading to mitochondrial dysfunction and eventually, the disease”.

Reviewer 2 Report

This MS by Jiang et al. explored the effects of Aβ species on the mitochondria function and cell survival in cardiac cells. It operated in both cell lines and primary cells as well as in mitochondria isolated from rat hearts. Overall, the experiments were designed carefully, and the conclusion sounds reasonable. There are some concerns as follows.

1) Did the author try lower levels of Aβ both Aβ1-40 and Aβ1-42, as 10μM is a rather high concentration?

2) For Aβ1-40 from Figure 2, the staining is far weaker than that for Ab1-42. Is it because Aβ1-40 is less internalized than Aβ1-42? The authors stated,  "Aβ1-40 showed more cytoplasmic signal, in addition to the nuclei." Did the authors mean Aβ distributes in both cytosol and nucleus? In either way, it is better to have magnified pictures. Also, how to avoid the background signals from endogenous Aβ?

3) Figure 6 has big problems. The resolution is too low. For Figure 6B, did Aβ1-40 reduce the levels of ATP5A as the signal was far weaker? The Aβ signal for Aβ1-42 looks not real but just non-specific binding. For Figure 6A, the Aβ signal for Aβ1-40 is even less than control? If Aβ1-40 is less internalized, it is hard to draw a conclusion. 

Round 2

Reviewer 2 Report

The author may replace Fig 6 with the so-called high-resolution image in Supplementary material. 

Author Response

We replaced Fig.6 with supplementary Fig.1S (TIFF file) as suggested by the reviewer. However, as we mentioned earlier, insertion of images (JPEG, TIFF, and any other files) in Word file affects the quality of the images by the Word platform. Thank you!